# The Fe-FeSi phase diagram at Mercury's core conditions

E. Edmund [1,2,8✉], G. Morard[1,3], M. A. Baron[1], A. Rivoldini [4], S. Yokoo [5], S. Boccato[1], K. Hirose [5,6], A. Pakhomova [7] & D. Antonangeli [1]

Mercury's metallic core is expected to have formed under highly reducing conditions, resulting in the presence of significant quantities of silicon alloyed to iron. Here we present the phase diagram of the Fe-FeSi system, reconstructed from in situ X-ray diffraction measurements at pressure and temperature conditions spanning over those expected for Mercury's core, and ex situ chemical analysis of recovered samples. Under high pressure, we do not observe a miscibility gap between the cubic fcc and B2 structures, but rather the formation of a re-entrant bcc phase at temperatures close to melting. Upon melting, the investigated alloys are observed to evolve towards two distinct Fe-rich and Fe-poor liquid compositions at pressures below 35-38 GPa. The evolution of the phase diagram with pressure and temperature prescribes a range of possible core crystallization regimes, with strong dependence on the Si abundance of the core.

[1] Sorbonne Université, Muséum National d'Histoire Naturelle, UMR CNRS 7590, Institut de Minéralogie, de Physique des Matériaux et de Cosmochimie, IMPMC, 75005 Paris, France. [2] Centre for High Pressure Science and Technology Advanced Research (HPSTAR), Shanghai 201203, China. [3] Université Grenoble Alpes, Université Savoie Mont Blanc, CNRS, IRD, Université Gustave-Eiffel, ISTerre, 38000 Grenoble, France. [4] Royal Observatory of Belgium, Brussels, Belgium. [5] Department of Earth and Planetary Science, The University of Tokyo, Tokyo, Japan. [6] Earth-Life Science Institute, Tokyo Institute of Technology, Meguro, Tokyo 152-8550, Japan. [7] Photon Science, Deutsches Elektronen-Synchrotron, D-22607 Hamburg, Germany. [8] Present address: Earth and Planets Laboratory, Carnegie Institution for Science, Washington, DC 20015, USA. ✉email: eedmund@carnegiescience.edu

The interior structure of Mercury provides a unique window into unusual planetary chemistry as well as insight into planetary diversity as a whole. Indeed, it is not uncommon that concepts which underpin our current understanding of a given planet (including the Earth) stem from ideas developed to describe observations of other telluric bodies (e.g., planetary differentiation via a deep magma ocean[1]). Among the telluric planets of the solar system, Mercury is the most reduced, with estimated oxygen fugacities between −2.6 and −7.3 log units below the Fe–FeO buffer[2–5]. At such reducing conditions it is expected that significant quantities of Si have partitioned into Mercury's core over the timescales of planetary differentiation, e.g.[5–8]. Furthermore, the large uncompressed density of Mercury requires the presence of a large metallic core. Thus, it is generally accepted for Mercury to have a Si-bearing core comprising more than 80% of the radius of the planet. Still, present estimates of the Si abundance of Mercury's core spans almost the entire Fe–FeSi phase diagram[7,9,10], with further complexity due to the possible presence of S or C[7,8,11,12]. Ongoing missions to Mercury such as BepiColombo[13] are expected to provide further constraints on its internal structure.

It is important to recognize, however, that all reliable compositional and dynamical models of planetary interiors cannot overlook the physical properties of candidate materials and need to make use of these properties to match geophysical observables. Consequently our knowledge of the interior structure, dynamics and composition of Mercury are fundamentally limited by the degree to which the structure and properties of Mercury-forming materials can be constrained at the relevant pressures and temperatures. As a consequence of Mercury's large core, existing at pressures of between about 5–40 GPa, and temperatures between about 1600–3000 K, i.e.[14], the bulk properties of the planet itself are primarily dictated by the thermal and chemical state of the core. In particular, the stable solid phases and their melting relations provide fundamental constraints not only on the density profile of Mercury, but also on the planet's heat budget[15–17] and core crystallization regime[18,19]. Understanding these features of Mercury, in turn, are required to model dynamo mechanisms within the core capable of explaining the planet's magnetic field, e.g.[20,21].

In spite of its importance to the interior of Mercury, there is considerable disagreement on the Fe–FeSi phase diagram and eutectic composition at the relevant P-T conditions[22–26], and many studies do not attempt to resolve the phase relations in this system at pressures below 50 GPa and at high temperatures, e.g.[27–29]. Indeed, the Fe–FeSi phase diagram displays considerable complexity at ambient pressure[30] and understanding the evolution from ambient-pressure complexity to Mbar-pressure simplicity[31] is critical for understanding the interior structure and dynamics of not only Mercury, but of telluric planets in general. As a matter of fact, there are numerous compositional models which are proposed to account for geodetic observations of Mercury while disagreeing strongly on the bulk composition of the planet[32], references therein. This lack of consensus necessitates thorough measurements on core-candidate alloys, in order to better understand the internal structure and dynamo mechanisms of Mercury.

To this end, the crystal structures of Fe–Si alloys spanning the entire Fe–FeSi phase diagram have been studied in situ at high P-T conditions using laser-heated diamond anvil cells (DACs) and synchrotron X-ray diffraction (XRD). These measurements have been complemented with ex situ analysis of recovered samples via focused ion beam (FIB) milling and textural mapping, and by chemical analysis employing scanning electron microscopy (SEM) in order to provide a clear description of both the structure and composition of these materials at the conditions of Mercury's core.

## Results and discussion

**Solid phases of the Fe–FeSi system.** At ambient pressure and temperature, Fe–Si solid solutions form a bcc structure (space group: Im-3m) with random placement of Si atoms in the Fe–Si unit cell for concentrations below 4 wt% Si. For larger quantities of Si, B2-type ordering (space group: Pm-3m) is observed at concentrations between 4 and 6 wt% Si, and DO$_3$-type ordering (space group: Fm-3m) is observed from about 5–17 wt% Si[33]. Elevated temperatures substantially increase the compositional stability field of bcc-type solid solutions with respect to ordered variants[30,34]. At even higher Si concentrations and high temperatures, Fe–Si alloys form two stoichiometric compounds— ($\beta$-)Fe$_2$Si (space group: P-3m1)[35] and (B20-) FeSi (space group: P213)[36]. Eutectic liquid compositions are located at about 19.2 wt% Si and 21.5 wt% Si above the DO$_3$ + $\beta$, and $\beta$ + B20 mixed phase regions, respectively[30]. While $\beta$-Fe$_2$Si is not known to exist at the high pressures relevant to planetary cores, e.g.[22,26,37], the presence or absence of solid-solid miscibility gaps (i.e., extended regions where two distinct coexisting solid phases are thermodynamically stable), and their evolution with pressure, temperature or composition represent crucial parameters in tracking both the liquid eutectic composition and the crystallization regimes of planetary cores. Literature on the Fe–Si system at high pressures report a series of miscibility gaps in the solid phase, between Si-poor hcp or fcc phases and an Si-rich B2 phase[26,28,38], or between the Si-rich B2 phase and stoichiometric B20 FeSi[22,26]. However, ex situ chemical analysis has only confirmed the existence of the hcp + B2 and B2 + B20 miscibility gaps at high pressures and temperatures, with the former only extending to solidus temperatures at pressures well beyond those of the core of Mercury (>40 GPa)[26,28,38]. The presence of an fcc + B2 miscibility gap which extends to solidus temperatures represents a possible invariant point in the Fe–Si phase diagram which can help to explain observations of increasing Si solubility in the eutectic liquid at low pressures, and the subsequent decrease in Si solubility observed at higher pressures, e.g.[29]. Furthermore, at present almost all DAC studies at high pressure on the Fe–Si system report liquid compositions which are bracketed to between 9 and 16 wt% Si at pressures between 40 and 80 GPa[26,29,39], differently from multi-anvil studies at lower pressures which have not observed an Fe-rich miscibility gap at high temperatures, and have reported a substantially more Si-rich liquid eutectic composition of 25.1 wt% Si[22] at 21 GPa, within the compositional range of the B2 + B20 miscibility gap. In order to reconcile the seemingly discrepant DAC and multi-anvil studies, substantial changes must occur to the Fe–Si phase diagram between 20 and 40 GPa to connect these contrasting observations.

In the present study, the investigated Fe–Si compositions, namely Fe5Si, Fe7Si, Fe16Si, Fe22Si, Fe28Si and Fe30Si (see "Methods", Supplementary Table 1) were synthesized by Physical Vapor Deposition (PVD), which leads to highly non-equilibrium structures, e.g.[33]. This substantially reduces the kinetic barriers for reaching equilibrium states at high P-T conditions. Furthermore the very low initial grain size of the starting materials (<100 nm) greatly enhances structural resolution of the solid phases present at very high temperatures by providing a larger time window before the heated alloy recrystallizes into a few large single crystals.

Figure 1a shows the results of a heating cycle on Fe7Si, where with increasing temperature, pure fcc phase is observed before the crystallization of a bcc phase just below melting.

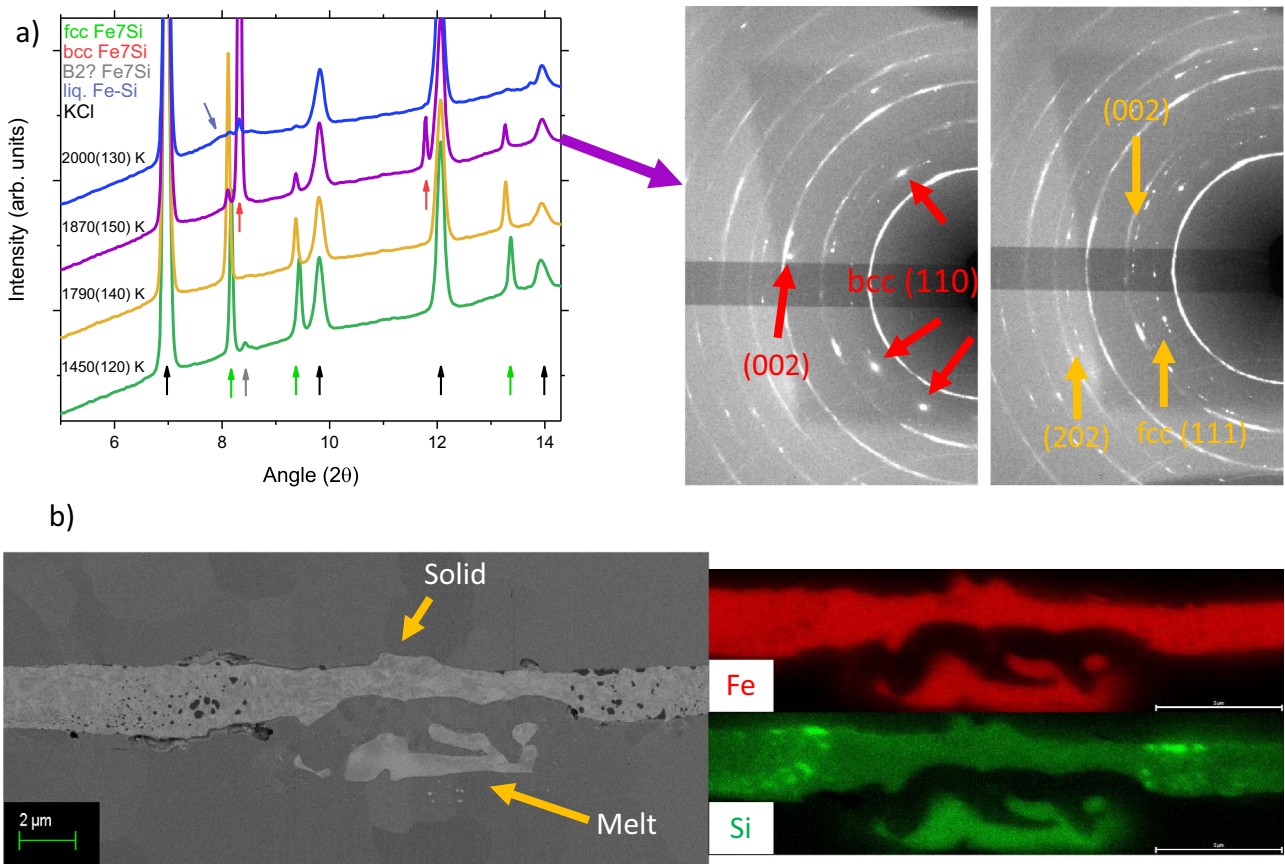

**Fig. 1 In situ X-ray diffraction and ex situ chemical analysis of Fe7Si. a** Integrated diffraction patterns of Fe7Si at 17 ± 2 GPa and 1450–2000 K showing a single-phase fcc structure, followed by a transition at higher temperatures to a bcc structure and then melting at the highest temperature. On the right side, diffraction images collected at 1870 K (left) and 1790 K (right) are shown, which indicate a clear change in the texture of the sample on transformation to the bcc phase and a significant loss of intensity of the fcc phase in the 1870 K image. **b** Textural map of Fe7Si (left), quenched from 34 ± 2 GPa and chemical analysis (right) of the hotspot showing clear enrichment of the liquid with Si, both solid and liquid compositions are homogeneous within error of the technique.

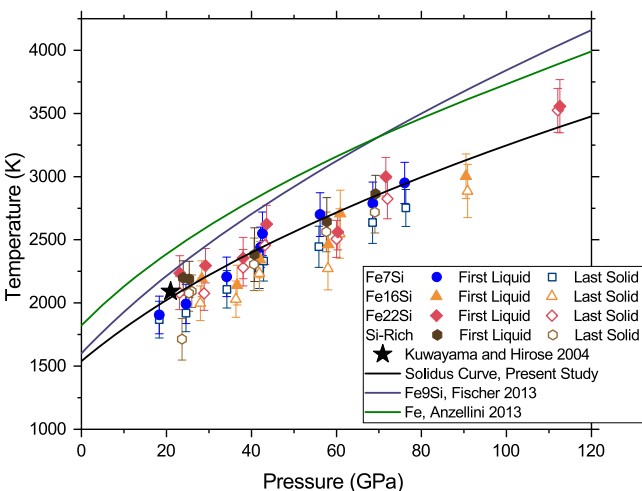

**Fig. 2 Melting temperatures in the Fe–FeSi system at high pressures.** Pressure-temperature points at which liquid diffuse scattering is first detected (solid symbols) and the last diffraction pattern prior (open symbols). Solid lines indicate estimated solidus temperatures of the Fe-Si system from the present study and available literature. Error bars in temperature represent one standard deviation. Solid black line shows a Simon–Glatzel fit to the present dataset with parameters $T_0 = 1538$ K, $a = 24.66$, $c = 2.17$[84].

These observations deeply modify the current vision of the Fe–Si system. It was suggested in previous studies that Si solubility is lower in the fcc phase than the hcp phase, e.g.[26,40] and thus the observation of an hcp + B2 region (observed at pressures above 23 GPa and low temperatures) would have implied the observation of an fcc + B2 region upon temperature increase. In the present study, the stability of a bcc phase is based on the precise determination of the volume of both the bcc and fcc structures, the observation of a single fcc phase for the Fe7Si composition at moderate temperatures, and the compositional constraints placed via chemical analysis of the recovered samples.

In contrast, previous studies of this region of P-T space solely determined phase equilibria using in situ XRD, reporting large regions with 3-phase mixtures, possibly due to differences in X-ray beam diameter relative to hotspot size[26,38]; for further discussion see Supplementary Discussion 1 and Supplementary Figs. 1–5. It is observed that the volume of the bcc phase is systematically larger than that of the fcc phase at similar pressures and temperatures across the measured P-T range of this study, indicative of an entropically-stabilized phase (see Supplementary Fig. 1).

The solid phases of Fe16Si, Fe22Si, and Fe30Si are mostly consistent with the phase relations and phase boundary P-T reported in literature. In brief, Fe16Si is observed to crystallize into the DO$_3$ structure for pressures below 38 GPa, but DO$_3$ reflections are lost at >2000 K likely due to thermal disordering of

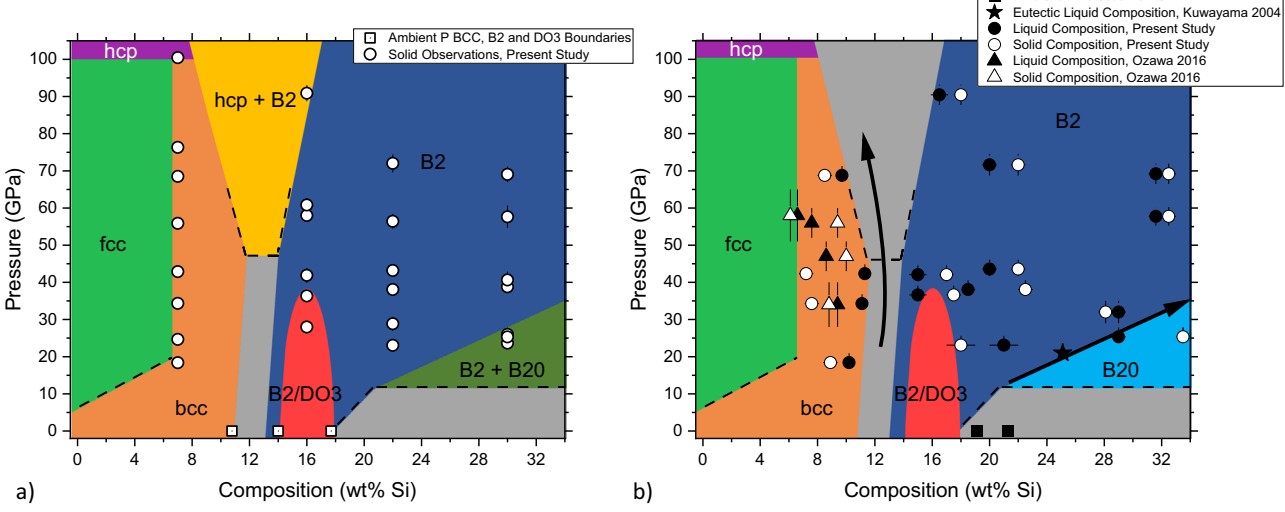

**Fig. 3 Solidus and liquidus projections for the Fe–Si system at high pressures. a** Pressure–composition (P-X) phase diagram showing subsolidus phase relations just below melting. Open circles indicate the P-X conditions of solid measurement, and open squares indicate ambient pressure solubility limits of the bcc, B2, and DO3 phases[30]. **b** Pressure–composition phase diagram showing the liquidus phases and eutectic points of the Fe-Si system. Open and closed circles indicate solid and liquid compositions of recovered samples from the present study, respectively. Open and closed triangles indicate solid and liquid compositions reported in Ozawa et al.[29]. Squares and the star indicate eutectic points from literature[22,30]. Error bars for both figures represent one standard deviation for pressure and composition. For further details see text.

the alloy at high temperatures, e.g.[41] (Supplementary Fig. 6), as the DO₃ structure is only stable in a very narrow compositional range at high temperatures.

Fe22Si is observed to be stable in the B2 structure over the entire pressure range investigated (Supplementary Fig. 7). For Fe30Si (Supplementary Fig. 8), decompression of the sample at high temperatures (1400–1600 K) lead to the observation of a single B2 phase at pressures above 27(1) GPa, and the formation of a B2 + B20 mixture at lower pressures.

**Melting relations and phase diagram of the Fe–FeSi system.** The melting relations and their variation across the different solidus phases of the Fe–FeSi system are of prime interest for understanding the influence of composition on inner core crystallization and dynamics. Figure 2 presents the solidus melting curve of the Fe–Si system up to 115 GPa and 3500 K from the first observation of liquid diffuse scattering. It is observed that regardless of the starting composition of the alloy, the onset of melting is within uncertainties, indicating that the melting temperature is only weakly dependent on composition over the studied alloys, but reduced relative to Fe[42,43]. Furthermore, low-pressure observations are in good agreement with experiments performed in multi-anvil apparatus with high precision thermocouples[22]. Comparison with previous DAC studies shows relatively good agreement (Supplementary Fig. 9) with the present solidus melting curve for alloys with high Si content (from 15 to 18 wt% Si)[39,44,45]. However, melting curves of alloys with lower Si contents disagree with the present study (9–10 wt% Si)[26,44] suggesting somewhat higher melting temperatures than those reported here (Fig. 2). This disagreement may be attributed to differences in melting criteria, as Fischer et al.[26] uses an average of liquid diffuse scattering and laser power vs. temperature plateaus to determine solidus temperatures, or due to the difficulties of producing sufficient melt to detect diffuse scattering near eutectic temperatures in earlier studies. When solely comparing the results of the present study to the observation of diffuse scattering in Fischer et al.[26], differences larger than experimental uncertainties only arise at pressures below 60 GPa. Moreover, similarities between the melting temperature of Fe7Si and Fe16Si

are to be expected also on the basis of the similarity in melt composition between these two samples—the quenched melt pools differ in composition by about 4 wt% Si at 50 GPa (Fig. 3).

Regardless, we conclude that Si alloying reduces melting temperatures of pure Fe, at the conditions of Mercury's core ($\Delta T \sim -240$ K at 25 GPa) and at those of Earth's core-mantle boundary (CMB) ($\Delta T \sim -530$ K at 135 GPa).

Figure 3 provides subsolidus and solidus phase relations as determined by in situ structural characterization and ex situ chemical analysis of quenched samples (also refer to Fig. 5a and Supplementary Figs. 10 and 11 for temperature–composition phase diagrams at selected pressures). At pressures below 35 GPa, the melting of Si-rich (>28 wt% Si at 34 GPa, >20 wt% Si at 23 GPa) samples leads to a eutectic composition that follows the disappearance of the B2 + B20 stability field. Based on available literature B20 FeSi is no longer thermodynamically stable at pressures above 30 or 40 GPa[24,26]. However, this compositional evolution toward an Si-rich alloy is not observed at any pressure for Fe16Si, which alongside Fe7Si, melts toward a composition bracketed to between 12 and 15 wt% Si at 37 ± 2 GPa. While this pressure is at the edge of the DO₃ stability field, the interpretation of a common Fe-rich melting minimum between Fe7Si and Fe16Si is supported by analysis of the volumes of the quenched Fe–Si melt at pressures down to 12 GPa (Supplementary Fig. 12). The bracketed composition for this feature of the Fe-rich side of the phase diagram spans the compositional stability field of the B2 and DO₃ phases at ambient pressure and solidus temperatures[30].

The evolution of the Fe–FeSi system from ambient pressure to those existing in the core of Mercury (Fig. 3, see also Fig. 5) sees the expansion of fcc and B2 stability fields, and the progressive disappearance of the bcc, DO₃ and B20 structures. It has been noted previously that at ambient or low pressures (1 GPa) the melting loop becomes very narrow in correspondence to the B2 and DO₃ structures. The difference between solidus and liquidus temperatures at constant composition becomes <20 K between 10 and 17 wt% Si, and varying composition across this region leads to changes of liquidus and solidus temperatures of <20 K per wt% Si[34,46]. Such behavior indicates that small changes to phase

stability and physical properties at these conditions can significantly alter the boundaries between these phases. The unusual stability of the $DO_3$ structure, which at high pressures transforms to B2 structures upon either decrease or increase of Si content, has been reported previously[26]. This can be related to the notion that stabilization of the $DO_3$ structure, the most ordered phase for the non-stoichiometric Fe–Si alloys, is due to the net effect of strong competing interatomic interactions and long range forces[47].

Recent studies have indicated that the elastic properties of bcc-structured Fe–Si alloys differ substantially from those of B2 or $DO_3$ alloys at similar compositions, while the elasticity of B2 and $DO_3$ phases are similar[33,39,48]. As melting minima in binary systems lie either above a solid solution or a miscibility gap, the present results are consistent with an azeotropic minimum over the B2/DO3 solid solutions or a eutectic over a miscibility gap between two of these 'bcc-like' phases. Both scenarios are found either in the Fe–Si phase diagram at ambient pressure[49], or related iron-based alloys, e.g., Fe–V,[35].

At higher pressures, the hcp + B2 region expands in temperature to cross the solidus above 50–70 GPa, leading to the decrease of eutectic liquid Si content reported by DAC experiments at Mbar pressures[26,29,50].

**The crystallization paradigms of Mercury's inner core.** Mercury's core is believed to be composed primarily of Fe, containing significant quantities of Si, but to approach more realistic core compositions one must also consider the possible presence of Ni, S and/or C. Modeling of the partitioning of siderophile elements during Mercury's differentiation indicate that the core likely contains between 2.6 and 7.1 wt% Ni[8]. At the conditions of Mercury's core, this quantity of nickel does not influence the melting temperature[51], and it leads to an expansion of fcc phase stability in the Fe–Ni–Si phase diagram while retaining the sequence of solid solutions observed in the Fe–Si binary[52,53]. Interestingly, while C may also play a prominent role in the formation of the core[8,54,55], the presence of Ni reduces the partitioning of C into solid Fe[56,57].

The solubility of sulfur in solid iron is low at the pressures and temperatures of Mercury's core[58]. As a consequence of these factors, the bulk crystallization regime of the core is primarily governed by the phase diagram of the Fe–Si system.

Importantly, in the absence of seismological data, our knowledge of the deep interior of Mercury is primarily a result of the influence of these phases on the interpretation of available geodetic information, magnetic field observations and surface chemistry, e.g.[7,14,21]. The temperature and pressure variation of an isentropic liquid core can be calculated by coupling constraints on core mass and radius ($R_C$), e.g.[14] with available liquid iron alloy equations of state[59–62], for a given CMB temperature.

Constraints can be placed based on the expected core temperature profile relative to the melting curves of Fe alloys and estimated CMB temperatures. The observation of ancient volcanic plains on the surface of Mercury[63], combined with the present day absence of active volcanism[64] indicates that the lower mantle of Mercury is currently near or below solidus temperatures, with current estimates ranging between 1800 and 2000 K depending on the assumed mineral composition[65,66]. While uncertainties remain over the composition and transport properties of the core-forming material, e.g.[67,68], Mercury's core is expected to have cooled by <200 K in the last 4.5 Gyr based on current estimates of planetary contraction[15–17,69].

Figure 4 shows the gradients of calculated Fe–Si isentropes (isentropic gradients) and Fe–Si melting relations. It can be observed that with the exception of the most Si-poor

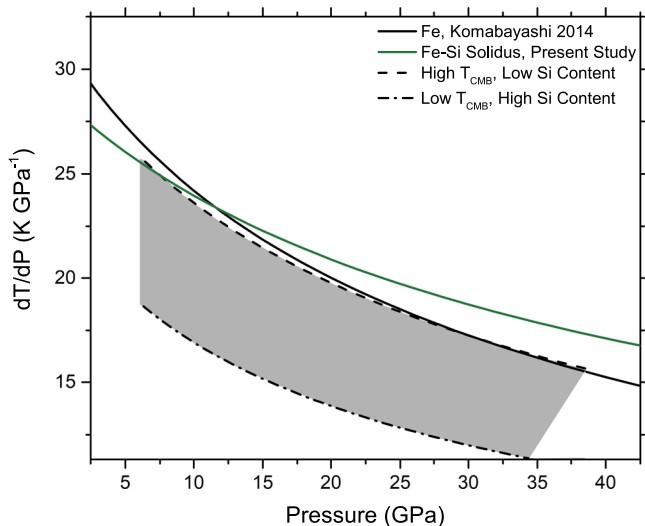

**Fig. 4 Melting curve gradients ($dT_m/dP$) and isentropic gradients ($dT_S/dP$) vs. pressure for the Fe–Si system.** Dashed lines and dash-dotted lines indicate the maximum range of $dT_S/dP$ compatible with compositional and thermodynamic constraints outlined in text. Solid lines show $dT_m/dP$ from the present study for the Fe–Si system, and from literature for Fe[59]. It is observed that the expected range of possible values for $dT_S/dP$ (shaded gray area) is systematically lower than $dT_m/dP$ for Fe (black) and in the Fe–Si system (green), indicating that an Fe–Si core would crystallize in a bottom-up scenario.

compositions, such a core would crystallize through a bottom-up scenario. While for Si-poor compositions the adiabat gradients follow closely the gradient of the melting of Fe, these isentropes require unreasonably high CMB temperatures (>2000 K) and lie above liquidus temperatures at all pressures. Such a scenario is incompatible with a dynamo driven by solid crystallization. Core compositions with more plausible temperature profiles range from 5 to 16 wt% Si for the Fe–Si system when varying core radius between 1950 and 2050 km (see Supplementary Fig. 13a). Further calculations incorporating the effect of S or C alloying indicate that these elements reduce the Si content of the core by roughly 2.0–2.3 wt% per wt% S, or 1.5–1.8 wt% per wt% C, for a given $T_{CMB}$ and core radius. For larger core radii the combined reduction of solidus and liquidus temperatures due to the incorporation of Si and S or C outpace the degree to which these elements modify isentropic temperature profiles (Supplementary Fig. 13b). Higher Si concentrations may yet be possible due to immiscibility at low pressures of liquid Fe–S and liquid Fe–Si, as a liquid Fe–S layer near the CMB would lead to depletion of iron in the Si-bearing liquid[70]. However, current estimates of sulfur partitioning between metal and silicates at the redox conditions of Mercury[5], alongside the immiscibility of C and S in liquid Fe at <6 GPa[71] necessitate further study to support such a scenario.

The complexity of the Fe–Si phase diagram at pertinent pressures and compositions leads to highly divergent scenarios for inner core crystallization and development, schematized in Fig. 5a, b alongside temperature–composition projections at various pressures from 0 to 40 GPa.

Si concentrations of about 7–12 wt% Si would lead to the formation of a solid inner core crystallizing in the bcc or B2 structure (the estimated boundary positions between these phases are indicated by dashed lines at 25 and 40 GPa in Fig. 5). The narrow P-T-X stability field of the fcc, bcc and B2 structures in this region of the Fe–Si phase diagram suggests that within this compositional range, it is likely to have structural layering within

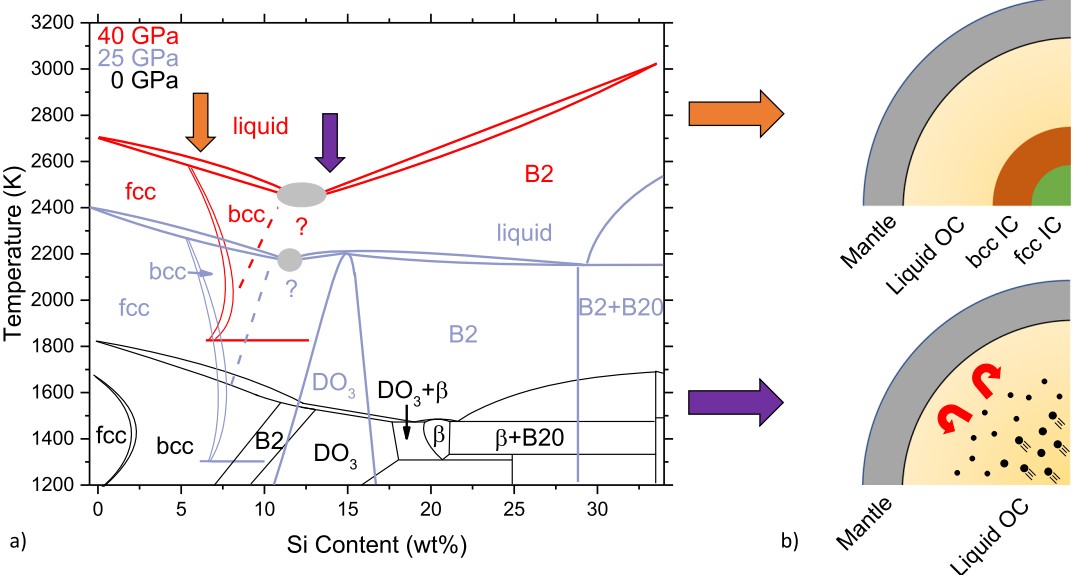

**Fig. 5 Temperature–composition phase diagram of the Fe–Si system and possible core crystallization regimes. a** The temperature–composition (T-X) phase diagram of the Fe–FeSi system at 25 and 40 GPa from the present study, and the ambient pressure phase diagram from literature[30,34,35]. Phase diagrams at 25 and 40 GPa with datapoints are shown in Supplementary Figs. 10 and 11. Gray circles and question marks indicate uncertainty over the nature of the invariant point. **b** Possible scenarios for the structure and crystallization regimes of Mercury for: (i) Si-poor (orange arrows) and (ii) Si-rich (purple arrows) liquid cores. (i) The crystallization of an Fe-rich bcc alloy at compositions near the fcc–bcc boundary can lead to the progressive crystallization of an innermost fcc core surrounded by a solid bcc shell. (ii) The crystallization of more Si-rich material would lead to formation of a solid which is less dense than its corresponding liquid, rising upwards and leading to convection without an inner solid core.

Mercury's inner core due to the progressive enrichment of Si in the outer core, coupled with the sharp T-X slopes of these phases. While some studies have invoked internal structural transitions between crystalline phases of iron at the conditions of Earth's inner core to explain seismological observations of depth-dependent anisotropy[72,73], the implications of such an effect for Mercury is not well understood and has not been reported previously. As metallurgical literature indicates that the fcc–bcc transition of Fe–Si alloys requires some degree of undercooling (typically between 10 and 40 K, Cockett and Davis[74]) and Mercury's core is expected to cool by about 30 K per Gyr[16,17] such a transition can occur progressively across geologic time and, due to the lower entropy of the fcc phase, may provide an additional heat source in present day Mercury.

Intermediate Si concentrations (about 12–16 wt% Si) would lead to crystallization of a buoyant solid within the core and the possibility of dynamo generation without the presence of a solid inner core. This scenario has been suggested for the Fe–FeS system, but it requires the presence of S in larger quantities than compatible with geochemical constraints at the redox conditions expected for Mercury's core[75]. For more Si-rich compositions (>16 wt% Si), the behavior is more complex. While solid Fe–Si would be buoyant for these alloys at pressures higher than those of the B20-B2 transition in stoichiometric FeSi, at pressures below the B2-B20 transition the crystallizing B2 Fe–Si solid would be denser than the resulting Fe–Si liquid. Current literature places this transition between 30 and 40 GPa[24,26], while here the pressure at the center of Mercury is estimated to be around 36–38 GPa in Fig. 4. Depending on the precise boundary and potential compositional effects (e.g., alloying with C, S, Ni), this transition and the resulting change in sign of solid/liquid buoyancy would lead to the formation of an Fe–Si 'cloud layer' within Mercury's core. In the case that the B20-B2 transition occurs at pressures beyond those found at the center of Mercury, this region of the phase diagram would lead to the formation of a B2 inner core. For even higher Si concentrations (>20–25 wt%),

the arrival of a liquid Fe–Si core to the composition of the B2 + B20 eutectic (25 wt% Si at 21 GPa[22]) would terminate the formation of an inner core and lead to an FeSi snow regime, providing a hard limit on the size of an Si-rich Fe–Si core. As B20 FeSi has melting temperatures comparable to pure iron[25], this phase would likely persist in the solid state to much lower pressures. However the compositions at which these scenarios occur are at the upper compositional bounds of our calculated models.

To summarize, the Fe–FeSi system has been thoroughly studied at high temperatures and pressures encompassing the conditions of Mercury's core. At these conditions this system exhibits up to four solid solutions (fcc, bcc, B2, and DO3), and two compositionally distinct melting minima are present. These observations establish a diverse array of crystallization regimes over a compositional range which was previously thought to either exhibit wide miscibility gaps between three phases[26], or a single extended solid solution[22]. The complexity of the Fe–Si system at Mercury's core conditions prescribes a core evolution which is sensitive to Si abundance, and presents possible new mechanisms of core evolution, such as the structural transition between different solid solutions upon cooling of the inner core, or the occurrence of an Fe–Si 'cloud layer' due to the change in sign of solid/liquid Si partitioning in B2-structured Si-rich Fe–Si alloys.

## Methods

**Sample synthesis and preparation**. The alloys investigated in this study were synthesized by PVD at l'Institut de Minéralogie, de Physique des Matériaux et de Cosmochimie (IMPMC). The sample compositions were measured using energy dispersive X-ray spectroscopy (EDX) in a SEM, and the estimated compositions of these alloys based on analysis of the recovered samples is reported in Supplementary Table 1.

Compositions of the starting materials were measured far from the hotspot, on unheated portions of the sample cross section. Variations in Si concentration were <0.3 wt% for Fe7Si, <0.5 wt% Si for Fe16Si and Fe22Si, and 2 wt% for Fe30Si as discussed below. For Fe5Si and Fe28Si, only one measurement was performed for each alloy, and so variance cannot be determined. For Fe30Si, due to the challenges

 

of synthesizing Si-rich, off-stoichiometric samples, the starting material was deliberately made chemically heterogeneous, varying from 26 to 33 wt% Si across the sample surface. Care was taken to load samples from a single area on the substrate to minimize compositional scatter, leading to variance of about 2 wt% from sample to sample as determined by EDX.

All starting materials were found to be either amorphous (Fe16Si, Fe22Si), nano-grained bcc alloys (Fe7Si) or a mixture of an amorphous and nano-grained alloy (Fe28Si, Fe30Si). Fe5Si was not measured by XRD. High-pressure experiments were performed with Le-Toullec-type membrane-driven DACs using diamonds with culets ranging from 350 down to 150 μm in diameter, equipped with 200 μm thick Re gaskets. The samples were mechanically etched from a glass substrate, and loaded in the DAC, sandwiched between disks of KCl (thicknesses of 10–20 μm depending on culet diameter). In order to remove the Fe–Si alloy from the glass substrate, a WC needle was used to create a scratch on the sample surface. This leads to the separation of the sample and substrate, which would propagate upwards of ~500 microns from the location of the mark. A tungsten needle was then used to break off a piece of the sample of the desired dimensions, as tungsten is not hard enough to damage the underlying substrate unless in direct contact. KCl disks acted as both thermal and chemical insulators between the sample and diamonds, in order to minimize temperature gradients and carbon diffusion into the bulk sample. To minimize moisture contamination the KCl/sample/KCl assembly was dried for several hours in a vacuum oven at 80 degrees C before measurement. Loaded sample thicknesses were typically between 2 and 4 μm.

**In situ sample characterization.** Synchrotron angle-dispersive XRD measurements were performed on beamline P02.2 at Petra III[76] for Fe7Si, Fe16Si, Fe22Si and Fe30Si. Monochromatic X-rays ($\lambda = 0.2901$ nm) were employed in transmission geometry, and focused to ~2 μm by 2 μm horizontal by vertical full-width-at-half-maximum (FWHM). Diffraction images were collected using a Perkin Elmer area detector with collection times varying from 1 to 5 s, and these images were radially integrated using the Dioptas software package[77] with the detector configuration calibrated by a $CeO_2$ standard. In order to generate high temperatures, double-sided off-axis laser heating was employed. A single Nd:YAG laser was split into two optical paths, and focused on the two opposite faces of the sample. Intensity of the laser beam impinging on the sample for each optical path was varied separately through the use of $\lambda/2$ waveplates. The size of the heating spot was ~20 by 20 μm $H \times V$ FWHM in diameter, much larger than the FWHM of the X-ray beam. Temperatures were measured on both optical paths by the spectroradiometric method. Reported temperatures were determined by a Planck fit to the observed blackbody radiation at the center of the heating spot, and corrected downwards by 3% to account for axial gradients across the diffracted volume of the sample[78]. Lattice parameters of the sample and KCl have been determined using the Le Bail method as implemented in Jana2006[79]. Pressure applied to the sample at both ambient and high temperatures has been estimated using the determined lattice parameters of KCl and the KCl P-V-T equation of state reported in Dewaele et al.[80]. The bulk temperature of KCl is estimated by Eq. (1) after Campbell et al.[78]:

$$T_{\text{KCl}} = (T_{\text{sample}} \times 3 + 300)/4 \qquad (1)$$

XRD patterns were collected with increasing temperature until melting, detected by the appearance of diffuse scattering from the sample, and then quenched. Off-line laser-heating runs were performed on sample Fe28Si. In these experiments, pressures were determined based on the measurement of the fluorescence of a ruby chip embedded in the sample chamber[81], or the Raman shift of the diamond anvil $T_{2g}$ phonon[82] before and after laser heating. These pressure values were corrected for the estimated thermal pressure of KCl (about 2 GPa). Pressure measured by ruby fluorescence or Raman spectroscopy of the diamond anvils before and after heating did not differ significantly relative to experimental uncertainties. Laser power was increased for 15–30 s until the sample reached the desired temperature, held for 3 s, and then quenched.

**Ex situ sample characterization.** Synchrotron run products (Fe7Si, Fe16Si, Fe22Si, Fe30Si) and off-line heating experiments (Fe5Si, Fe28Si) were analyzed texturally and chemically via FIB and SEM. In order to do so, the culet sections of the Re gasket containing the KCl/sample/KCl assembly were cut out of the bulk gasket with a picosecond laser at l'Institut de Physique du Globe de Paris. The Re disks containing the samples were oriented on a glass slide for an initial rough polishing using Ar ion milling, to expose the heated spots of the sample prior to FIB milling and further characterization. Further details of FIB procedures are reported elsewhere[83]. After FIB milling to the desired depth in the sample, Fe and Si abundances were determined using EDX. This was performed using a Zeiss Ultra-55 Field Emission Gun SEM (IMPMC, Paris), with an electron beam smaller than 1 μm and operation voltage of 15 keV. Emitted intensity was collected on a silicon drift detector, with intensities calibrated against a copper standard. To analyze the samples and quantify composition, the exposed heating spots were coated with 3 nm of Pt, and Fe and Si abundance were calibrated against Fe (for Fe) and FeSi (for Si) standards. This calibration was cross-checked against $Fe_5Si_3$ and Si, all coated with the same thickness of Pt. Repeated measurements on FeSi ($N = 15$) generated variations in Si content of <0.3 at.% Si (~0.15 wt% Si), however it was observed that for intermediate compounds between Fe and Si ($Fe_5Si_3$, FeSi

and $FeSi_2$) differences between Si and FeSi as the Si standard resulted in changes of ~0.4–0.7 at.% Si in composition, representing the systematic error in measurement due to the choice of calibration.

## Data availability

Data presented in this article can be found in Supplementary Dataset 1.

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

## Acknowledgements

This project has received funding from the European Research Council (ERC) under the European Union's Horizon 2020 Research and Innovation Programme (grant agreement No. 724690). M.A.B. has received funding from the European Research Council (ERC) under the European Union Horizon 2020 Research and Innovation Programme (grant agreement No. 670787). PVD machines at IMPMC have been supported by the European Research Council (ERC) under the European Union Horizon 2020 Research and Innovation Programme (grant agreement No. 670787). The Focused Ion Beam and Scanning Electron Microscope facilities at IMPMC are supported by Région Ile-de-France grant SESAME 2006 No. I-07-593/R, INSU-CNRS, Institut de Physique (INP)-CNRS, Université Pierre et Marie Curie-

ARTICLE

Paris 6, and by the French National Research Agency (ANR) grant ANR-07-BLAN-0124-01. We acknowledge DESY (Hamburg, Germany), a member of the Helmholtz Association HGF, for the provision of experimental facilities. Parts of this research were carried out at P02.2 beamline of Petra III. The work of A. Rivoldini was financially supported by the Belgian PRODEX program managed by the European Space Agency in collaboration with the Belgian Federal Science Policy Office. The authors wish to thank A. Boury for sample synthesis at IMPMC. The authors wish to thank I. Esteve and B. Doisneau for assistance with SEM and FIB measurements. The authors wish to thank J. Badro and N. Weir at Institut du Paris du Globe (IPGP) for picosecond laser machining. The authors wish to thank R. Husband and N. Giordano for temperature calibration testing on Petra III beamline P02.2.

## Author contributions

E.E., G.M., and D.A. conceived the experiments. E.E., G.M., M.A.B., S.Y., S.B., A.P., and D.A. performed the experiments. K.H. and D.A. provided resources for experiments. E.E., M.A.B., S.B., and S.Y. analyzed the data. A.R. performed thermodynamic calculations and interior structure modeling. E.E. produced the figures and wrote the manuscript, with input from all other co-authors.

## Competing interests

The authors declare no competing interests.

## Additional information

ns license, unless indicated otherwise in a credit line to the material. If material is not included in the article's Creative Commons license and your intended use is not permitted by statutory regulation or exceeds the permitted use, you will need to obtain permission directly from the copyright holder. To view a copy of this license, visit http://creativecommons.org/licenses/by/4.0/.

© The Author(s) 2022

9