## [Peer Review File · Nature Communications]

REVIEWER COMMENTS

Reviewer #1 (Remarks to the Author):

This manuscript by Edmund et al. is an experimental investigation of the Fe-FeSi phase diagram under high pressure and temperature, with an application to Mercury's core. Constraining the composition and structure of Mercury's core is fundamental to understand core cooling processes in the innermost planet of our Solar System, and in particular, to explain the weak intensity of the intrinsic magnetic field. The present-day state of Mercury's core is still largely unknown – for instance, there is no evidence for a solid inner core, though its presence is predicted by several studies. Regarding its composition, the highly reduced conditions of Mercury imply that a large amount of Si should be present in the metallic core (together with some S), but the stability of phases over the large pressure range spanned by Mercury's core is not well defined yet and requires further investigation. With the BepiColombo mission en route to Mercury, new studies of the planet's interior like the one by Edmund and coauthors are timely and will likely help interpret upcoming field observations and data.

This experimental work by Edmund et al. places new and high-quality constraints on the Fe-FeSi phase diagram, using 6 different compositions. Interestingly, their results point out some differences with the melting curve of Fe-Si alloys from previous studies. The authors propose some plausible explanations for this discrepancy, related to how the presence of melt is determined using diffraction data. In the last part of the paper, the experimental results are applied to the core of Mercury. As detailed below, this study would represent - after some revisions - a substantial improvement of our knowledge of the Hermean core.

The main comment about this manuscript regards its structure -- it would benefit from expanding the application to Mercury's core, because this is ultimately the focus of the study, as indicated by the title of the paper. However, the current text is essentially dedicated to the Fe-FeSi phase diagram, corresponding to a total of about 7 pages (sections 2 and 3), whereas the application to Mercury occupies less than 3 pages (section 5). This comes as imbalanced, and it makes the discussion of the paper more technical than it probably should. Especially for the high-impact journal that is targeted here, the emphasis should be placed on the application to the planet (i.e., the "big picture" message) rather than specific aspects of a phase diagram.

Developing and strengthening Section 5 would allow addressing the following points:

- 1) It is pointed out l. 264-265 that the core of Mercury is primarily composed of Fe, Si and S. What about Ni? Meteorite geochemistry studies have suggested that nickel is abundant in metallic cores, and recent experimental works under pressure have shown that the presence of Ni strongly influences the melting behavior of the Fe-FeS system as well as some transport properties. Should the same be expected for the Fe-FeSi system? Can the effect of Ni on the phase diagram be ignored here?
- 2) Crystallization in Mercury's core is addressed very succinctly by presenting different scenarios that are mostly based on phase transformations and density considerations. However, a clear comparison with Mercury's thermal state is missing, and this really needs to be added. Several modeling studies have proposed adiabats for Mercury's core through planetary cooling, and it would be interesting to compare these adiabats with the liquid determined in the present study, especially their intersection (because it would indicate the onset of crystallization). For instance, a quick comparison between the estimated T CMB for present-day Mercury and the phase diagram at 25 GPa in Figure 3 suggests that an Fe-Si outer core would be fully liquid. What about deeper in the core? Will these comparisons between models and experiments agree with existing field observations? A comparison of the authors' results with existing models of the thermal state of Mercury's core would significantly improve the discussion.
- 3) Related to point 2) above, the possibility of a complex snow regime is highlighted in the conclusion (l. 340), but this point is not addressed clearly in the discussion (snow zone is only briefly invoked l. 322). It is definitely an important and relevant topic, and it would deserve more attention in this paper.

Other comments:

- What is the pressure range considered for this study? Figure 1 in Methods show data from 15 to 100 GPa (and in Figure 6, up to 112 GPa), but the T-X phase diagrams are only shown for P=25 and 40 GPa (probably because these two pressures bracket Mercury's present-day core). It would be interesting and relevant here to see how the phase diagram evolves until 100 GPa, and the experiments are here!

- Introduction: a paragraph mentioning the effect of Si on the physical properties of iron alloys (velocity, density, thermal and electrical conductivities) should be added. These properties are actually relevant to interpret field observations (magnetic data, surface chemistry data, geodetic constraints) in terms of inner structure.
- L. 327: "while Si may not be known to strongly influence liquid properties": can the authors be more specific? Si definitely affects strongly some transport properties!
- L. 128: the Fe₅Si composition is not listed though it appears to have been investigated (see Methods section).
- The Methods section mentions SEM analyses several times in order to determine the chemical composition of phases – where are these results? (they are not shown in any table).
- Figure 2b is really difficult to read (and the garish colors really do not help!). It needs some improvement.
- Some typos need to be corrected (e.g., "criteria" l. 204).

I hope these comments are helpful to the authors as part of their revisions.

Reviewer #2 (Remarks to the Author):

In my opinion, this manuscript tells an interesting story, uses good figures, and extremely well-written text. I learned a lot about the Fe-Si phase diagram and Mercury's core by reading this manuscript.

But I do have one major critique — it is not clear why the phase diagrams proposed here should be trusted in the areas that they are in contradiction with the results of [Fischer et al. 2013 EPSL 373 (2013) 54–64]. The main areas of disagreement seem to be (A) the bcc phase at ~ 7wt% Si in the region just below melting, and (B) the melting curve at ~ 9 wt%Si. I think that it is the responsibility of this new article to either make a very strong case of why the slightly different methods used here are more reliable, or more simply, to show much, much more data in the supplementary material.

To make a very strong case for improved reliability, I would am unlikely to be convinced by common arguments in mineral physics papers about the laser spot size being large compared to the X-ray beam, or smaller steps in temperature during heating. Rather, I would want to see evidence that the hotspot was actually wider (e.g. photos) and a sequence of temperature fits to show that control was actually better in this study when the X-rays detected the bcc phase, as compared to the hotspot than in the Fischer et al. when the X-rays did not detect the bcc phase. I acknowledge that the starting material used here might be crucial for easing the synthesis of the bcc phase, but it isn't clear to me that the starting material's advantage (small grain size and/or amorphous material) outweighs its disadvantage (more oxidized surface).

Alternatively, I would be satisfied by a very long supplement that shows stacks of integrated X-ray patterns for all the data showing a transition from fcc to bcc to melting for Fe₇Si. There are eight data points for bcc at 17 to 100 GPa in Fig. 2b, so I would expect at least eight heating runs to be shown. If possible, I would want to see reversals from melting to bcc. If no such reversals were detected, I recommend that this be stated clearly. I realize this is not the standard for mineral physics publications, but I think it should be, at least in cases where a new paper contradicts an older paper that used very similar techniques. Otherwise, published mineral physics data sets will continue to accumulate with contradictory information and no way to resolve the discrepancies.

A related technical issue is that I did not understand -- What is the difference between bcc (Im-3m) with random substitutions of Si into pure Fe, and B2 (Fm-3m) with Fe substituted for some of the Si atoms in FeSi? I'm unsure about the atomic positions, and I'm unsure how the X-ray data in this manuscript discriminate between bcc and B2.

As far as "suitability for Nature Communications", I am ambivalent so far. On the one hand, the topic is interesting, the writing is outstanding, including the scenarios about Mercury's core history, and a lot of data and experimental effort went into this project. On the other hand, I am not yet convinced that the bcc phase is reproducibly and unambiguously detected in this study, which would cast doubt on the interesting scenario shown by the orange arrow in Fig. 3. Since I don't know the Mercury literature very well, I am not sure if the scenario of the purple arrow in Fig. 3 is a new idea.

Finally, here are some minor suggestions to consider:

- Line 206: Delete "or"
- Fig. 2: Add labels (a), (b), (c)
- Lines 301-306: Please make this statement about a heat source more explicit. My guess is that bcc transforms to fcc upon cooling, which releases heat because the bcc phase has higher entropy.
- Methods Line 16: Please explain how you "mechanically etch" samples from a glass substrate. Do you use a razor blade, a plastic tool, a WC tool?
- Supplement Fig. 2: Add "Neither" to make the phrase "neither melting nor..."

Reviewer #3 (Remarks to the Author):

Overall comment: This study reports experimental results on the Fe-Si system over a range of P, T, and X for which there is very little data. The authors make an excellent connection to this P-T-X regime and the conditions under which Mercury's core formed and possibly partially crystallized. The experiments and corresponding data all appear to be of a high quality, and the results have important implications for the state of Mercury's core, particularly for the solid inner core. My only criticism with this work is that the links to Mercury could be made a bit stronger by contextualizing the results within the framework of existing literature about the composition of Mercury's core. The manuscript does not consider carbon as a light element, and it does not adequately address the wide range of estimates for the Si content of Mercury's core on the basis of both geochemical and geophysical observations. I have provided a few detailed comments below to aid the authors in making such connections, and I think that this work will be highly suitable for publication after the connections to Mercury are made stronger. As far as the relevance of this work to Nature Communications, these results are broadly applicable across the Earth and planetary sciences as well as materials science. Furthermore, this study provides important new constraints on a fundamental metallic system. Consequently, I think this work is sufficiently impactful to warrant publication in Nature Communications.

General Comment: Although I appreciate the efforts to keep manuscripts concise, I think this manuscript would benefit with a bit more information on the constraints and estimates for Si abundances of Mercury's core in the introduction and latter parts of the discussion to better contextualize the experimental results and the conclusions. At present, there is a fairly wide range of estimates for the Si content of Mercury's core (<1-25 wt.% Si) both from geochemical and geophysical observations and/or constraints (Chabot et al., 2014; Hauck II et al., 2013; Margot et al., 2019; Nittler et al., 2018; Vander Kaaden et al., 2020). The experimental results here add valuable knowledge that was previously missing, but the authors should make some additional efforts to connect their results to what we think we know (and don't know) about Mercury and the Si content of its core.

Lines 27-28: The range of estimates for Mercury's oxygen fugacity is a bit wider, with a range of ΔIW -2.6 to -7.3 (McCubbin et al., 2012; McCubbin et al., 2017; Namur et al., 2016; Zolotov et al., 2013). This range is important because it may have implications for the abundance of Si in Mercury's core given the reaction $2FeO + Si \rightleftharpoons SiO_2 + 2Fe$, which is dependent on fO_2 .

Lines 33-37: Carbon has also been a big part of the discussion as a possible light element in Mercury's core along with S and Si (Li et al., 2015; Vander Kaaden et al., 2020). The case for C comes in part from the idea of a potential graphite flotation crust on Mercury that may indicate a C-saturated core (Klima et al., 2018; Peplowski et al., 2016; Vander Kaaden and McCubbin, 2015). Please be sure to include C as part of the discussion to at least the same degree you discuss S.

Lines 280-282: After reading the Terasaki et al (2019) manuscript cited here. I do not see where the compositional constraint of 6 wt% Si in Mercury's core comes from unless you have fixed or assumed the amount of S. Please provide some additional details and/or caveats. Furthermore, this constraint (i.e., ≥ 6 wt.% Si) will almost certainly change if a wider variety of light elements are considered in Mercury's core, like carbon.

Lines 48-61 in SI: Please provide additional details on the technique, beam conditions, and spot sizes used for the compositional measurements. I presume you used energy dispersive spectroscopy to quantify the abundances of Fe and Si, not scanning electron microscopy.

Cited references:

Chabot, N.L., Wollack, E.A., Klima, R.L., Minitti, M.E., 2014. Experimental constraints on Mercury's core composition. *Earth and Planetary Science Letters* 390, 199-208.

Hauck II, S.A., Margot, J.-L., Solomon, S.C., Phillips, R.J., Johnson, C.L., Lemoine, F.G., Mazarico, E., McCoy, T.J., Padovan, S., Peale, S.J., Perry, M.E., Smith, D.E., Zuber, M.T., 2013. The curious case of Mercury's internal structure. *Journal of Geophysical Research: Planets* 118, 1204-1220.

Klima, R.L., Denevi, B.W., Ernst, C.M., Murchie, S.L., Peplowski, P.N., 2018. Global Distribution and Spectral Properties of Low-Reflectance Material on Mercury. *Geophysical Research Letters* 45, 2945-2953.

Li, Y., Dasgupta, R., Tsuno, K., 2015. The effects of sulfur, silicon, water, and oxygen fugacity on carbon solubility and partitioning in Fe-rich alloy and silicate melt systems at 3 GPa and 1600 degrees C: Implications for core-mantle differentiation and degassing of magma oceans and reduced planetary mantles. *Earth and Planetary Science Letters* 415, 54-66.

Margot, J.-L., Hauck II, S.A., Mazarico, E., Padovan, S., Peale, S.J., 2019. Mercury's Internal Structure, in: Solomon, S.C., Nittler, L.R., Anderson, B.J. (Eds.), *Mercury: The View after MESSENGER*. Cambridge University Press, pp. 85-113.

McCubbin, F.M., Riner, M.A., Vander Kaaden, K.E., Burkemper, L.K., 2012. Is Mercury a volatile-rich planet? *Geophysical Research Letters* 39.

McCubbin, F.M., Vander Kaaden, K.E., Peplowski, P.N., Bell, A.S., Nittler, L.R., Boyce, J.W., Evans, L.G., Keller, L.P., Elardo, S.M., McCoy, T.J., 2017. A Low O/Si Ratio on the Surface of Mercury: Evidence for Silicon Smelting? *Journal of Geophysical Research: Planets* 122, 2053-2076.

Namur, O., Charlier, B., Holtz, F., Cartier, C., McCammon, C., 2016. Sulfur solubility in reduced mafic silicate melts: Implications for the speciation and distribution of sulfur on Mercury. *Earth and Planetary Science Letters* 448, 102-114.

Nittler, L.R., Chabot, N.L., Grove, T.L., Peplowski, P.N., 2018. The Chemical Composition of Mercury, in: Solomon, S.C., Nittler, L.R., Anderson, B.J. (Eds.), *Mercury: The View after MESSENGER*. Cambridge University Press, pp. 30-51.

Peplowski, P.N., Klima, R.L., Lawrence, D.J., Ernst, C.M., Denevi, B.W., Frank, E.A., Goldsten, J.O., Murchie, S.L., Nittler, L.R., Solomon, S.C., 2016. Remote sensing evidence for an ancient carbon-bearing crust on Mercury. *Nature Geoscience* 9, 273-276.

Vander Kaaden, K.E., McCubbin, F.M., 2015. Exotic crust formation on Mercury: Consequences of a shallow, FeO-poor mantle. *Journal of Geophysical Research-Planets* 120, 195-209.

Vander Kaaden, K.E., McCubbin, F.M., Turner, A.A., Ross, D.K., 2020. Constraints on the Abundances of Carbon and Silicon in Mercury's Core From Experiments in the Fe-Si-C System. *Journal of Geophysical Research: Planets* 125, e2019JE006239.

Zolotov, M.Y., Sprague, A.L., Hauck, S.A., Nittler, L.R., Solomon, S.C., Weider, S.Z., 2013. The redox state, FeO content, and origin of sulfur-rich magmas on Mercury. *Journal of Geophysical Research-Planets* 118, 138-146.

Dear Reviewers,

Thank you for your detailed comments, criticisms and suggestions regarding the present manuscript - the response to which we present below, in conjunction with our submission of the revised manuscript. All updated text in the article has been bolded for ease of identification. Particular adjustments to figures will be discussed in the body of the reviewer response (relevant for Figures 2-4).

Reviewer #1 (Remarks to the Author):

This manuscript by Edmund et al. is an experimental investigation of the Fe-FeSi phase diagram under high pressure and temperature, with an application to Mercury's core. Constraining the composition and structure of Mercury's core is fundamental to understand core cooling processes in the innermost planet of our Solar System, and in particular, to explain the weak intensity of the intrinsic magnetic field. The present-day state of Mercury's core is still largely unknown – for instance, there is no evidence for a solid inner core, though its presence is predicted by several studies. Regarding its composition, the highly reduced conditions of Mercury imply that a large amount of Si should be present in the metallic core (together with some S), but the stability of phases over the large pressure range spanned by Mercury's core is not well defined yet and requires further investigation. With the BepiColombo mission en route to Mercury, new studies of the planet's interior like the one by Edmund and coauthors are timely and will likely help interpret upcoming field observations and data.

This experimental work by Edmund et al. places new and high-quality constraints on the Fe-FeSi phase diagram, using 6 different compositions. Interestingly, their results point out some differences with the melting curve of Fe-Si alloys from previous studies. The authors propose some plausible explanations for this discrepancy, related to how the presence of melt is determined using diffraction data. In the last part of the paper, the experimental results are applied to the core of Mercury. As detailed below, this study would represent - after some revisions - a substantial improvement of our knowledge of the Hermean core.

The main comment about this manuscript regards its structure -- it would benefit from expanding the application to Mercury's core, because this is ultimately the focus of the study, as indicated by the title of the paper. However, the current text is essentially dedicated to the Fe-FeSi phase diagram, corresponding to a total of about 7 pages (sections 2 and 3), whereas the application to Mercury occupies less than 3 pages (section 5). This comes as imbalanced, and it makes the discussion of the paper more technical than it probably should. Especially for the high-impact journal that is targeted here, the emphasis should be placed on the application to the planet (i.e., the “big picture” message) rather than specific aspects of a phase diagram.

Thank you for your comments. Alongside the comments of Reviewer #3, section 5 has been expanded and some technical aspects of the phase diagram have been moved into the supplemental information. Particular points made will be addressed below.

Developing and strengthening Section 5 would allow addressing the following points:

1) It is pointed out l. 264-265 that the core of Mercury is primarily composed of Fe, Si and S. What about Ni? Meteorite geochemistry studies have suggested that nickel is abundant in metallic cores, and recent experimental works under pressure have shown that the presence of Ni strongly

influences the melting behavior of the Fe-FeS system as well as some transport properties. Should the same be expected for the Fe-FeSi system? Can the effect of Ni on the phase diagram be ignored here?

A more systematic discussion of other alloying elements has been incorporated into Section 5 - see lines 261-280 of the main text for justification regarding the importance of the Fe-FeSi system.

In particular, we have chosen not to discuss the transport properties of Fe and Fe-alloys, as values of thermal and electrical conductivity are controversial, even across different groups using nominally similar techniques, and do not represent the focus of this work. As well, alternative constraints on thermal history are available based on estimates of planetary contraction. For both of these points and specific references see lines 299-305.

2) Crystallization in Mercury's core is addressed very succinctly by presenting different scenarios that are mostly based on phase transformations and density considerations. However, a clear comparison with Mercury's thermal state is missing, and this really needs to be added. Several modeling studies have proposed adiabats for Mercury's core through planetary cooling, and it would be interesting to compare these adiabats with the liquid determined in the present study, especially their intersection (because it would indicate the onset of crystallization). For instance, a quick comparison between the estimated T CMB for present-day Mercury and the phase diagram at 25 GPa in Figure 3 suggests that an Fe-Si outer core would be fully liquid. What about deeper in the core? Will these comparisons between models and experiments agree with existing field observations? A comparison of the authors' results with existing models of the thermal state of Mercury's core would significantly improve the discussion.

Point 2 has been addressed by expanding on discussion of the melting curves of Fe(C/S)-Si relative to the effect of these elements on isentrope temperatures – see Figure 4, Supplemental Figures 13a, and b, lines 306-324 in the main text, in order to provide a clearer picture of the importance Fe-FeSi phase diagram in determining the chemical evolution of Mercury's core. It is worth noting that we do not compare to existing thermal models, as such models depend heavily on the thermal EoS of the liquid phases, and much of the experimental data for Fe-alloys has been published in the last few years (c.f. Terasaki et al., 2019, Morard et al., 2017, Xu et al., 2021)). Our calculations here use the latest data and thermal EoS.

Where applicable, we compare the present dataset to our own calculations, however we have deliberately chosen to avoid discussion of how specific aspects of the phase diagram relate to field observations, as many aspects of this phase diagram relate to fundamentally new phenomena with no established analogue among the telluric planets, or within modelling literature (for example, the possibility of internal structural transitions within a solid inner core as a heat source – lines 340-356, novel dynamics due to the change of sign of solid/liquid partitioning of Si-rich B2 alloys - lines 363-373).

3) Related to point 2) above, the possibility of a complex snow regime is highlighted in the conclusion (l. 340), but this point is not addressed clearly in the discussion (snow zone is only briefly invoked l. 322). It is definitely an important and relevant topic, and it would deserve more attention in this paper.

Text relating to snow regimes has been expanded in lines 373-385.

Other comments:

- What is the pressure range considered for this study? Figure 1 in Methods show data from 15 to 100 GPa (and in Figure 6, up to 112 GPa), but the T-X phase diagrams are only shown for P=25 and 40 GPa (probably because these two pressures bracket Mercury's present-day core). It would be interesting and relevant here to see how the phase diagram evolves until 100 GPa, and the experiments are here!

Given the complexity of the Fe-FeSi phase diagram at the P-T conditions of Mercury's core, we prefer to focus our discussion on this region of the phase diagram. Given the higher granularity of the data at 100 GPa, and that most of the current controversy of the Mbar-pressure phase diagram occurs at pressures beyond this (120-150 GPa, Ozawa 2016 & Fischer 2013), we feel that this is better suited to future studies.

- Introduction: a paragraph mentioning the effect of Si on the physical properties of iron alloys (velocity, density, thermal and electrical conductivities) should be added. These properties are actually relevant to interpret field observations (magnetic data, surface chemistry data, geodetic constraints) in terms of inner structure.

Due to the breadth of discussion and interpretation required for both our reported phase diagram and model calculations, we feel that it is better to maintain a strong focus on the phase diagram of the Fe-FeSi system as the main point of the paper.

- L. 327: "while Si may not be known to strongly influence liquid properties": can the authors be more specific? Si definitely affects strongly some transport properties!

This sentence has been reworked for clarity.

- L. 128: the Fe₅Si composition is not listed though it appears to have been investigated (see Methods section).

This has been added.

- The Methods section mentions SEM analyses several times in order to determine the chemical composition of phases – where are these results? (they are not shown in any table).

SEM results have been incorporated into a supplemental data file.

- Figure 2b is really difficult to read (and the garish colors really do not help!). It needs some improvement.

Figure 2 has been separated into two separate figures, and the visual presentation of what was previously defined as 2b and 2c (now Figure 3) has been improved. As an aside, Figure 3 has been modified slightly such that the left and right panels are subsolidus and liquidus projections respectively, to minimize confusion.

- Some typos need to be corrected (e.g., "criteria" l. 204).

This has been updated.

I hope these comments are helpful to the authors as part of their revisions.

We thank the reviewer for these constructive comments.

Reviewer #2 (Remarks to the Author):

In my opinion, this manuscript tells an interesting story, uses good figures, and extremely well-written text. I learned a lot about the Fe-Si phase diagram and Mercury's core by reading this manuscript.

But I do have one major critique — it is not clear why the phase diagrams proposed here should be trusted in the areas that they are in contradiction with the results of [Fischer et al. 2013 EPSL 373 (2013) 54–64]. The main areas of disagreement seem to be (A) the bcc phase at $\sim 7\text{wt}\%$ Si in the region just below melting, and (B) the melting curve at $\sim 9\text{wt}\%$ Si. I think that it is the responsibility of this new article to either make a very strong case of why the slightly different methods used here are more reliable, or more simply, to show much, much more data in the supplementary material.

To make a very strong case for improved reliability, I would am unlikely to be convinced by common arguments in mineral physics papers about the laser spot size being large compared to the X-ray beam, or smaller steps in temperature during heating. Rather, I would want to see evidence that the hotspot was actually wider (e.g. photos) and a sequence of temperature fits to show that control was actually better in this study when the X-rays detected the bcc phase, as compared to the hotspot than in the Fischer et al. when the X-rays did not detect the bcc phase. I acknowledge that the starting material used here might be crucial for easing the synthesis of the bcc phase, but it isn't clear to me that the starting material's advantage (small grain size and/or amorphous material) outweighs its disadvantage (more oxidized surface).

Alternatively, I would be satisfied by a very long supplement that shows stacks of integrated X-ray patterns for all the data showing a transition from fcc to bcc to melting for Fe₇Si. There are eight data points for bcc at 17 to 100 GPa in Fig. 2b, so I would expect at least eight heating runs to be shown. If possible, I would want to see reversals from melting to bcc. If no such reversals were detected, I recommend that this be stated clearly. I realize this is not the standard for mineral physics publications, but I think it should be, at least in cases where a new paper contradicts an older paper that used very similar techniques. Otherwise, published mineral physics data sets will continue to accumulate with contradictory information and no way to resolve the discrepancies.

Thank you for your comments and criticisms, we have duly addressed them below and in the revised manuscript. In order to address these comments we have moved some discussion of the bcc phase into the SI, and have incorporated more figures to clarify this discovery, alongside its observation in the context of contemporary literature. Two important aspects of this study which differ from Fischer et al., 2013, are: (i) that this previous study performed only in situ diffraction, whereas our study incorporates extensive chemical analysis of recovered samples; (ii) the use of nanograined alloys with regular thickness allows for substantially improved determination of sample volume at high P-T (as evidenced by Supplemental Figure S1), which was also instrumental to the determination of this phase.

A related technical issue is that I did not understand -- What is the difference between bcc (Im-3m) with random substitutions of Si into pure Fe, and B2 (Fm-3m) with Fe substituted for some of the Si atoms in FeSi? I'm unsure about the atomic positions, and I'm unsure how the X-ray data in this manuscript discriminate between bcc and B2.

A discussion of the differences between the ordered and disordered structures has been included in the Supplemental information in lines 72-78. We never observe superstructure reflections for this phase, in contrast with other alloys and quenched liquid. However it is important to note that this is not the sole discriminant for this phase in our study, in contrast to previous studies, as clear trends are also observed for sample volumes (see Supplemental Figure 1).

As far as "suitability for Nature Communications", I am ambivalent so far. On the one hand, the topic is interesting, the writing is outstanding, including the scenarios about Mercury's core history, and a lot of data and experimental effort went into this project. On the other hand, I am not yet convinced that the bcc phase is reproducibly and unambiguously detected in this study, which would cast doubt on the interesting scenario shown by the orange arrow in Fig. 3. Since I don't know the Mercury literature very well, I am not sure if the scenario of the purple arrow in Fig. 3 is a new idea.

Nearly all literature models of Mercury follow the solubility limits reported in Kuwayama & Hirose 2004, i.e. a continuous solid solution stable from 0-25 wt% of Si. The presence of a buoyant solidus phase between 12-15wt% Si would change substantially the expected crystallization scenario, and this range of Si-concentrations is central to current compositional estimates of the core (shown by our models in Supplementary Figure S13a). The novelty of the crystallization regimes discussed in this paper has been emphasized further in the text (lines 344-350 and lines 359-363).

As addressed in our reply to a previous point, the stability of the bcc phase is a firm conclusion, now more extensively presented and argued.

Finally, here are some minor suggestions to consider:

- Line 206: Delete "or"

This has been amended.

- Fig. 2: Add labels (a), (b), (c)

Figure 2 has been split into two separate figures, and updated for scientific clarity.

- Lines 301-306: Please make this statement about a heat source more explicit. My guess is that bcc transforms to fcc upon cooling, which releases heat because the bcc phase has higher entropy.

This has been added in lines 354-356.

- Methods Line 16: Please explain how you "mechanically etch" samples from a glass substrate. Do you use a razor blade, a plastic tool, a WC tool?

We have included further explanation of this part of the methods

- Supplement Fig. 2: Add "Neither" to make the phrase "neither melting nor..."

This has been amended.

Reviewer #3 (Remarks to the Author):

Overall comment: This study reports experimental results on the Fe-Si system over a range of P, T, and X for which there is very little data. The authors make an excellent connection to this P-T-X regime and the conditions under which Mercury's core formed and possibly partially crystallized. The experiments and corresponding data all appear to be of a high quality, and the results have important implications for the state of Mercury's core, particularly for the solid inner core. My only criticism with this work is that the links to Mercury could be made a bit stronger by contextualizing the results within the framework of existing literature about the composition of Mercury's core. The manuscript does not consider carbon as a light element, and it does not adequately address the wide range of estimates for the Si content of Mercury's core on the basis of both geochemical and geophysical observations. I have provided a few detailed comments below to aid the authors in making such connections, and I think that this work will be highly suitable for publication after the connections to Mercury are made stronger. As far as the relevance of this work to Nature Communications, these results are broadly applicable across the Earth and planetary sciences as well as materials science. Furthermore, this study provides important new constraints on a fundamental metallic system. Consequently, I think this work is sufficiently impactful to warrant publication in Nature Communications.

General Comment: Although I appreciate the efforts to keep manuscripts concise, I think this manuscript would benefit with a bit more information on the constraints and estimates for Si abundances of Mercury's core in the introduction and latter parts of the discussion to better contextualize the experimental results and the conclusions. At present, there is a fairly wide range of estimates for the Si content of Mercury's core (<1-25 wt.% Si) both from geochemical and geophysical observations and/or constraints (Chabot et al., 2014; Hauck II et al., 2013; Margot et al., 2019; Nittler et al., 2018; Vander Kaaden et al., 2020). The experimental results here add valuable knowledge that was previously missing, but the authors should make some additional efforts to connect their results to what we think we know (and don't know) about Mercury and the Si content of its core.

Thank you for your comments and suggestions. Discussion of Carbon as an alloying element has been incorporated into the Introduction in lines 38-40, Section 5 in lines 261-276, lines 317-324, lines 328-332, and Supplemental figures S13a and b. Clarification of the importance of the Fe-FeSi phase diagram to Mercury's core has been added based on available literature and our own calculations in section 5.

While indeed the full range of estimates for Si content ranges between <1-25 wt% Si, our calculations show that the extreme ends of these values (close to Fe or close to FeSi) are unlikely without core sizes deviating considerably from estimates post-MESSENGER, which typically range between 2000-2060 km (c.f. Genova et al., 2019, Steinbrugge et al., 2021). This can potentially be modified by the incorporation of large quantities of S or C, but then leads to issues regarding bulk core chemistry, core and CMB temperatures, and issues relating to available compositional constraints, discussed section 5.

Lines 27-28: The range of estimates for Mercury's oxygen fugacity is a bit wider, with a range of ΔIW -2.6 to -7.3 (McCubbin et al., 2012; McCubbin et al., 2017; Namur et al., 2016; Zolotov et al., 2013). This range is important because it may have implications for the abundance of Si in Mercury's core given the reaction $2FeO + Si \leftrightarrow SiO_2 + 2Fe$, which is dependent on fO_2 .

This has been updated.

Lines 33-37: Carbon has also been a big part of the discussion as a possible light element in Mercury's core along with S and Si (Li et al., 2015; Vander Kaaden et al., 2020). The case for C comes in part from the idea of a potential graphite flotation crust on Mercury that may indicate a C-saturated core (Klima et al., 2018; Peplowski et al., 2016; Vander Kaaden and McCubbin, 2015). Please be sure to include C as part of the discussion to at least the same degree you discuss S.

As mentioned above, discussion of C as an alloying element has been incorporated into the Introduction and section 5. Our calculations, which have been expanded to incorporate the effect of C alloying based on available experimental data (Morard et al., 2017b, Shimoyama et al., 2016 in the main text).

Lines 280-282: After reading the Terasaki et al (2019) manuscript cited here. I do not see where the compositional constraint of 6 wt% Si in Mercury's core comes from unless you have fixed or assumed the amount of S. Please provide some additional details and/or caveats. Furthermore, this constraint (i.e., ≥ 6 wt.% Si) will almost certainly change if a wider variety of light elements are considered in Mercury's core, like carbon.

In the expanded section 5 the bounds have been recalculated, and further details have been added about both the calculations and the effect of C/S.

Lines 48-61 in SI: Please provide additional details on the technique, beam conditions, and spot sizes used for the compositional measurements. I presume you used energy dispersive spectroscopy to quantify the abundances of Fe and Si, not scanning electron microscopy.

Further details have been added to the methods section, see SI lines 58-64. 'SEM' has been replaced by 'EDX' where applicable.

Cited references:

Chabot, N.L., Wollack, E.A., Klima, R.L., Minitti, M.E., 2014. Experimental constraints on Mercury's core composition. *Earth and Planetary Science Letters* 390, 199-208.

Hauck II, S.A., Margot, J.-L., Solomon, S.C., Phillips, R.J., Johnson, C.L., Lemoine, F.G., Mazarico, E., McCoy, T.J., Padovan, S., Peale, S.J., Perry, M.E., Smith, D.E., Zuber, M.T., 2013. The curious case of Mercury's internal structure. *Journal of Geophysical Research: Planets* 118, 1204-1220.

Klima, R.L., Denevi, B.W., Ernst, C.M., Murchie, S.L., Peplowski, P.N., 2018. Global Distribution and Spectral Properties of Low-Reflectance Material on Mercury. *Geophysical Research Letters* 45, 2945-2953.

Li, Y., Dasgupta, R., Tsuno, K., 2015. The effects of sulfur, silicon, water, and oxygen fugacity on carbon solubility and partitioning in Fe-rich alloy and silicate melt systems at 3 GPa and 1600

degrees C: Implications for core-mantle differentiation and degassing of magma oceans and reduced planetary mantles. *Earth and Planetary Science Letters* 415, 54-66.

Margot, J.-L., Hauck II, S.A., Mazarico, E., Padovan, S., Peale, S.J., 2019. Mercury's Internal Structure, in: Solomon, S.C., Nittler, L.R., Anderson, B.J. (Eds.), *Mercury: The View after MESSENGER*. Cambridge University Press, pp. 85-113.

McCubbin, F.M., Riner, M.A., Vander Kaaden, K.E., Burkemper, L.K., 2012. Is Mercury a volatile-rich planet? *Geophysical Research Letters* 39.

McCubbin, F.M., Vander Kaaden, K.E., Peplowski, P.N., Bell, A.S., Nittler, L.R., Boyce, J.W., Evans, L.G., Keller, L.P., Elardo, S.M., McCoy, T.J., 2017. A Low O/Si Ratio on the Surface of Mercury: Evidence for Silicon Smelting? *Journal of Geophysical Research: Planets* 122, 2053-2076.

Namur, O., Charlier, B., Holtz, F., Cartier, C., McCammon, C., 2016. Sulfur solubility in reduced mafic silicate melts: Implications for the speciation and distribution of sulfur on Mercury. *Earth and Planetary Science Letters* 448, 102-114.

Nittler, L.R., Chabot, N.L., Grove, T.L., Peplowski, P.N., 2018. The Chemical Composition of Mercury, in: Solomon, S.C., Nittler, L.R., Anderson, B.J. (Eds.), *Mercury: The View after MESSENGER*. Cambridge University Press, pp. 30-51.

Peplowski, P.N., Klima, R.L., Lawrence, D.J., Ernst, C.M., Denevi, B.W., Frank, E.A., Goldsten, J.O., Murchie, S.L., Nittler, L.R., Solomon, S.C., 2016. Remote sensing evidence for an ancient carbon-bearing crust on Mercury. *Nature Geoscience* 9, 273-276.

Vander Kaaden, K.E., McCubbin, F.M., 2015. Exotic crust formation on Mercury: Consequences of a shallow, FeO-poor mantle. *Journal of Geophysical Research-Planets* 120, 195-209.

Vander Kaaden, K.E., McCubbin, F.M., Turner, A.A., Ross, D.K., 2020. Constraints on the Abundances of Carbon and Silicon in Mercury's Core From Experiments in the Fe-Si-C System. *Journal of Geophysical Research: Planets* 125, e2019JE006239.

Zolotov, M.Y., Sprague, A.L., Hauck, S.A., Nittler, L.R., Solomon, S.C., Weider, S.Z., 2013. The redox state, FeO content, and origin of sulfur-rich magmas on Mercury. *Journal of Geophysical Research-Planets* 118, 138-146.

REVIEWERS' COMMENTS

Reviewer #4 (Remarks to the Author):

Review of: "The Fe-FeSi phase diagram: New crystallization Paradigms for Mercury's Core" by Edmund et al.

The manuscript is very well-written and the methods and data collection are clearly outlined. The figures are very clear and previous work is generally well-referenced i.e. mostly supported by the appropriate the studies. The data looks sound and is certainly of interest for the community. I actually have only very few comments, it is a very nice and interesting dataset and I congratulate the authors with his. However, the only concern I have is that I am slightly worried that the submitted manuscript is too specific/technical (the possible ranges in phase abundances is a bit confusing at times, and obviously complex). In addition, it seems that a very large number of scenarios can be reconciled with the data and cannot be excluded either. Perhaps the authors can more precisely/briefly indicate the relevant "high-impact" implications of their results? Again, the data looks great, but I think maybe more can be done concerning the description of the implications of it in a more general form, so that other people in the field can also appreciate the data to the fullest extent.

Line-by-line comments

Introduction

Lines 28-33 and/or line 38: Please add other studies (Chabot et al, 2014, EPSL; Steenstra and van Westrenen, 2020, Icarus) who specifically (experimentally) investigated this aspect for a wide range of mercury bulk compositions and core formation scenarios.

Line 68: "Enormous" is maybe a bit over-the-top. Replace with "significant" or considerable something similar.

Line 263: Again, core compositions for Mercury are discussed but the relevant studies that were specifically focused on this aspect should then be acknowledged. For example, Steenstra and van Westrenen (2020) modeled the distribution of Ni and especially C (and importantly Si too) in detail during differentiation of Mercury and these results should be discussed here and in other relevant segments of the manuscript too.

Line 281-285: On a side note, siderophile modeling studies, especially in conjunction with incorporation of different assumed end-member bulk Mercury compositions, can also provide key constraints on core compositions and core mass.

Reviewer #4 (Remarks to the Author):

Review of: "The Fe-FeSi phase diagram: New crystallization Paradigms for Mercury's Core" by Edmund et al.

The manuscript is very well-written and the methods and data collection are clearly outlined. The figures are very clear and previous work is generally well-referenced i.e. mostly supported by the appropriate the studies. The data looks sound and is certainly of interest for the community. I actually have only very few comments, it is a very nice and interesting dataset and I congratulate the authors with his.

Thank you for your appreciation of the quality of our work. Please find attached our response to requested revisions below, following relevant sections of the review.

However, the only concern I have is that I am slightly worried that the submitted manuscript is too specific/technical (the possible ranges in phase abundances is a bit confusing at times, and obviously complex). In addition, it seems that a very large number of scenarios can be reconciled with the data and cannot be excluded either.

The balance between experimental aspects, the physics of the Fe-Si system and the consequences for our understanding of Mars' core is the outcome of feedbacks, sometime slightly opposite, of several reviewers. We believe the current version well highlight the novelty and impact of our conclusion, without loosing ground on the experiments and observations that are at the basis of these conclusions. In particular, it is important to note that while many scenarios are compatible with currently available compositional constraints, some of these scenarios represent new mechanisms of dynamo generation, here proposed for the first time and for which there are no current terrestrial analogues. Consequently their manifestation in terms of physical observables (and whether these interior expressions would match observed behaviour) is completely unknown. Performing the calculations needed to assess whether one given scenario is more suitable for Mercury over another is beyond the scope of the present paper. The more detailed and specific studies of the thermoelastic properties of liquid Fe-Si (+S/C) alloys that are needed to provide tighter constraints are ongoing.

Perhaps the authors can more precisely/briefly indicate the relevant "high-impact" implications of their results? Again, the data looks great, but I think maybe more can be done concerning the description of the implications of it in a more general form, so that other people in the field can also appreciate the data to the fullest extent.

The conclusions section of the paper has been re-written and incorporated into the end of the discussion section to more effectively summarize the main results of the present study and their broad implications.

Line-by-line comments

Introduction

Lines 28-33 and/or line 38: Please add other studies (Chabot et al, 2014, EPSL; Steenstra and van Westrenen, 2020, Icarus) who specifically (experimentally) investigated this aspect for a wide range of mercury bulk compositions and core formation scenarios.

This has been added.

Line 68: "Enormous" is maybe a bit over-the-top. Replace with "significant" or considerable something similar.

This has been amended.

Line 263: Again, core compositions for Mercury are discussed but the relevant studies that were specifically focused on this aspect should then be acknowledged. For example, Steenstra and van Westrenen (2020) modeled the distribution of Ni and especially C (and importantly Si too) in detail during differentiation of Mercury and these results should be discussed here and in other relevant segments of the manuscript too.

Thank you for pointing this out, this sentence (lines 255-258) has been modified.

Line 281-285: On a side note, siderophile modeling studies, especially in conjunction with incorporation of different assumed end-member bulk Mercury compositions, can also provide key constraints on core compositions and core mass.

Lines 281-285 (now 275-281) are intended as a lead-in to calculations of the thermal structure of Mercury and relevant Fe-alloy compositions. This section has been adjusted for clarity.